# Characterization of Cover Crop Rooting Types from Integration of Rhizobox Imaging and Root Atlas Information

**DOI:** 10.3390/plants8110514

**Published:** 2019-11-17

**Authors:** Gernot Bodner, Willibald Loiskandl, Wilfried Hartl, Eva Erhart, Monika Sobotik

**Affiliations:** 1Institute of Agronomy, Department of Crop Sciences, University of Natural Resources and Life Sciences, Konrad-Lorenz-Straße 24, A-3430 Tulln, Austria; 2Austrian Society of Root Research, Muthgasse 18, A-1190 Vienna, Austria; willibald.loiskandl@boku.ac.at (W.L.); w.hartl@bioforschung.at (W.H.); e.erhart@bioforschung.at (E.E.); monika.sobotik@aon.at (M.S.); 3Institute for Soil Physics and Rural Water Management, University of Natural Resources and Life Sciences, Muthgasse 18, A-1190 Vienna, Austria; 4Bioforschung Austria, Esslinger Hauptstrasse 132-134, A-1220 Vienna, Austria; 5Pflanzensoziologisches Institut, Pichlern 9, A-4822 Bad Goisern, Austria

**Keywords:** root classification, image analysis, root phenotyping, cover crop species, root functions

## Abstract

Plant root systems are essential for sustainable agriculture, conveying resource-efficient genotypes and species with benefits to soil ecosystem functions. Targeted selection of species/genotypes depends on available root system information. Currently there is no standardized approach for comprehensive root system characterization, suggesting the need for data integration across methods and sources. Here, we combine field measured root descriptors from the classical Root Atlas series with traits from controlled-environment root imaging for 10 cover crop species to (i) detect descriptors scaling between distant experimental methods, (ii) provide traits for species classification, and (iii) discuss implications for cover crop ecosystem functions. Results revealed relation of single axes measures from root imaging (convex hull, primary-lateral length ratio) to Root Atlas field descriptors (depth, branching order). Using composite root variables (principal components) for branching, morphology, and assimilate investment traits, cover crops were classified into species with (i) topsoil-allocated large diameter rooting type, (ii) low-branched primary/shoot-born axes-dominated rooting type, and (iii) highly branched dense rooting type, with classification trait-dependent distinction according to depth distribution. Data integration facilitated identification of root classification variables to derive root-related cover crop distinction, indicating their agro-ecological functions.

## 1. Introduction

Plant root systems have evolved from simple rhizome-derived structures (closely linked to fungal symbionts) towards complex branched systems [1]. Ontologically they developed in response to various environmental signals and interact with a highly diverse microbiome [2]. 

Several approaches have been proposed to classify species as well as varieties within a species in relation to their rooting type, for example, anatomical origin of roots was used to distinguish plants as allorhizal (dicotyledons) versus homorhizal (monocotyledons) [3]. On the basis of root axes origin, a taxonomy for root system characterization was suggested, comprising a primary root system (embryonic radicle, basal roots emerging from the hypocotyl/mesocotyl, and their respective laterals) and a secondary root system (mainly in monocotyledons) composed of shoot-borne axes, emerging from stem nodes upon tillering, and their laterals [4]. This approach has been substantiated by insights into the distinct genetic control of different root axes types (e.g., for maize [5]). Fitter [6] used a topological characterization of root systems with herringbone and dichotomous branching as extreme types. With a focus on branching as major grouping criterion, Fitter’s classification has a strong functional focus related to resource uptake strategies [7,8]. Assimilate investment, for example, specific root length and tissue density, also provide important criteria for root classification in relation to environmental adaptation [9]. In an agronomic context, assimilate investment is of functional importance to identify high-yield compatible root architecture ideotypes [10]. 

Depending on their focus, classification systems rely on different root traits. Currently there is no standardized measurement approach providing comprehensive root datasets applicable beyond single classification concepts. Historic pioneers of root research relied on laborious field excavations to describe root systems under natural growing conditions ([11]; Root Atlas Series 1960–2009). The traditional quantitative methods mostly used destructive field sampling/coring and subsequent separation of roots from soil to obtain morphological traits such as root length and its depth distribution (for an overview see [12]). More recently, numerous novel methods have been developed in the context of plant phenotyping, comprising high throughput seedling root screens [13], two dimensional RGB and (hyper)spectral rhizobox imaging [14,15], and high resolution three dimensional methods (magnetic resonance imaging, X-ray computed tompgraphy [16,17]).

Although root phenotyping is quickly advancing with a higher degree of standardization [18,19], species-specific root–environment interactions are too complex to realistically expect a full understanding of root systems from a single experimental setup. Plants growing in natural field environments typically face multiple stresses driving their spatio-temporal pattern of root growth and architectural adaptations. Relevant traits thus largely depend on the environmental settings at the target environment [20]. Considering inter-annual changes in climatic growth factors and spatial variability of edaphic conditions, even within a constraint region, a single phenotyping experiment hardly represents the variety of environmental triggers determining the plasticity of root system architecture and might thus fail to obtain a meaningful classification of distinctive species/genotypes. Therefore, beyond development of improved measurement approaches, integration of different methods and data sources is essential to obtain a more comprehensive characterization on root systems.

Cover cropping has become a common agro-environmental practice to improve sustainability in agriculture, reducing soil and nutrient losses by erosion and leaching, increasing organic matter inputs, and ameliorating soil physical properties (aggregate stability, water infiltrability, aeration; see recent reviews [21,22,23]). Most agro-environmental functions of cover crops are tightly linked to their root system (e.g., nitrate uptake from deep layers against leaching [24], aggregate stability and macroporosity against erosion [25], root organic matter input [26]). Targeted selection of cover crops and their composition in mixtures therefore requires an appropriate species description including their root systems, which is currently largely lacking.

The specific objectives of this study were (i) to investigate relations between cover crop root system descriptors from the classical Root Atlas series with a standardized rhizobox root imaging system, (ii) to determine key root traits for a comprehensive species characterization and scaling of traits between single axis and field observations, and (iii) to derive distinctive cover crop rooting types to facilitate targeted species selection for optimizing agro-environmental impacts of cover cropping. The overall aim is to advance integration of diverse root datasets towards more comprehensive species/genotype root system characterization and, thereby, improve inference from single designed experiments towards variable (field) environments.

## 2. Results

### 2.1. Rhizobox-Based Root System Characterization

Rhizobox imaging has been used as procedure for evaluation of root system parameters under standardized conditions. Table 1 shows the morphological parameters obtained from image analysis (WinRhizo) of visible roots at the transparent observation window. Architectural parameters (Root System Analyzer, RootNav) related to the branching pattern of a single (30 cm; starting from the tip) axis are given in Table 2.

Total visible root length was highest in *Raphanus sativa*, which had also the highest proportion of roots in lower layers. Generally, deep root allocation was less distinctive among species (coefficient of variation (CV) 0.53) compared to total root length (CV 0.77).

Species with visible root length >2000 cm (corresponding to a visible density of > 0.67 cm cm^−2^ surface) were *R. sativus, Camelina sativa, Helianthus annuus, Phacelia tanacetifolia*, and *Vicia sativa*. The four non-legumes also allocated more than 10% root length into deep soil layers between 70 and 100 cm. Among the species with high total root length, *V. sativa* had a shallower distribution with only 7% in deep layers. *Trifolium alexandrinum* and *Vicia faba* were at the lower end of root length, statistically overlapping with *Avena strigosa* and *Linum usitatissimum*. From these species with low root length, *A. strigosa* was the only species allocating more than 10% of root length into deeper layers.

Average diameter of visible root axes (length distribution over diameter; Appendix A) was lowest in *L. usitatissimum* and highest in *H. annuus*. *L. usitatissimum* also had the highest proportion of root length <0.4 mm, whereas species with lowest fine root length were *V. faba* and *A. strigosa* (statistically similar to *C. sativa*, *V. sativa*, and *H. annuus*). 

Architectural characterization aimed to capture the branching pattern of laterals from a main axis. The distance between lateral branches in the branching zone was highest for *V. faba*, *V. sativa*, and *L. usitatissimum* (1.3 mm). The smallest distance between emerging laterals was found for *R. sativus* and *C. sativa* (0.2 mm). On average, lateral roots emerged from the lower order primary axes with an angle of 73.6°. The steepest emergence angle was found for *T. alexandrinum* and *V. sativa* (< 70°), whereas laterals of *A. strigosa* and *L. usitatissimum* were near to horizontal emergence (average 85.1°).

The ratio between the length of the parent root and its laterals was highly variable (CV 1.03), ranging from strong lateral dominance (e.g., *R. sativus, Fagopyrum esculentum*) to less branched species with root length dominated by primary axes (e.g., *A. strigosa, L. usitatissimum*).

The convex hull area, a descriptor for soil exploration, was highest for *P. tanacetifolia* (with similar values for *R. sativus* and *H annuus*). Species with reduced lateral branching also showed the lowest convex hull (*A. strigosa, L. usitatissimum*).

### 2.2. Root System Characterization from Root Atlas Data

Quantitative species descriptors were extracted from the Root Atlas series (Table 3) by (i) scanning hand drawings of in-situ excavations (root length) and (ii) using parameters indicated by the authors in their text (maximum depth, highest branching order of laterals).

Average root length of the excavated root axes was 2153.6 cm with a maximum in *H. annuus* and a minimum in *C. sativa* (CV 0.62). Maximum depth ranged from 60 to 180 cm and followed a similar ranking as total length. The highest branching order of lateral roots was indicated for *F. esculentum* (fifth order) and lowest for *V. sativa* and oat (third order).

### 2.3. Inter-Trait Relations

Relations among the single root descriptors from two data sources were tested to infer on (i) mutual relations among single descriptors across observation scales (single axis to field), and (ii) generalizability beyond the given experimental system (i.e., field excavation vs. rhizobox). Results are shown in Figure 1.

Within the different groups of traits (single axis architecture, visible root morphology, Root Atlas), most mutual relations were found between the architectural descriptors (three significant correlations; convex hull with lateral distance and primary-lateral ratio; lateral distance with primary-lateral ratio). The group of morphological descriptors had two significant mutual correlations (length and top-bottom ratio; diameter and fine root ratio) and Root Atlas-derived descriptors showed one correlation (length with depth). 

Across observation scales (single axis—whole visible root system) and experimental systems (rhizobox—Root Atlas), most relations with other parameters were found for lateral distance and convex hull. In particular, the convex hull of a single branched axis scaled significantly to both whole visible root traits as well as field descriptors. Lateral distance also scaled to visible root traits (length and top-to-bottom ratio), but not to field observations from the Root Atlas series. 

At the scale of visible rhizobox root traits, species with low distance between laterals tended to constitute large root systems (total length) with high allocation of axes also towards deeper soil layers. These larger sized root systems corresponded to field observed plants with highest rooting depth. 

### 2.4. Trait-Based Root Classification of Species

Following the idea of a data-driven classification of cover crop species according to their root characteristics, principal component analysis was applied to first derive comprehensive root descriptors and then cluster species according to similar rooting types. Three different parameter sets were used to compare classification results: (i) Root Atlas data only (nine species), (ii) rhizobox-descriptors only (10 species), and (ii) selected lower-level (more constitutive) rhizobox and Root Atlas descriptors. (For the rational of descriptors integrated into the respective classification, see: Materials and Methods).

Figure 2 shows the biplots (left panel) with trait vectors and species allocated according to their respective principal component (PC) scores. For Root Atlas data alone (Figure 2A), with three trait descriptors, two PCs were retained. In case of the rhizobox-only and the combined rhizobox and Root Atlas datasets, traits were captured by three latent variables (PCs) explaining 98.3% of the variance, with each composite variable representing distinct rooting features. 

The biplot from Root Atlas (Figure 2A) traits allocated length and maximum depth to PC1 (as also suggested from their mutual correlation, Figure 1), whereas the branching descriptor of maximum lateral order constituted the second grouping variable. 

For the rhizobox-only traits (Figure 2B), PC1 contained single axis architectural features (convex hull, angle of lateral root emergence, ratio of primary to laterals); PC2 was related to both morphological and architectural traits determining root system density (length, lateral distance) and depth distribution (top-to-bottom length ratio); and PC3 captured fineness of root axes (proportion of fine roots, average diameter).

Focusing on lower-level root system traits and including the respective data from the Root Atlas series (Figure 2C), the following variables were represented in the single PCs: in PC1, branching pattern (distance of laterals and maximum order); in PC2, distribution (angle and top-to-bottom ratio); and in PC3, fineness (proportion of fine axis). Maximum depth was not uniquely related to one single component—it entered PC2 (positively) and PC3 (negatively). 

On the basis of their PC-scores, the cover crop species were clustered according to similarities in their rooting pattern (Figure 2D–F; similarity measure: semi-partial *R*^2^ (sp*R*^2^)). 

With only three quantitative descriptors available from the Root Atlas series (Figure 2D; *T. alexandrinum* not described), similarity was suggested between size (length, depth)-dominated species (*R. sativus* and *H. annuus*), species distinguished by high branching order (*F. esculentum* and *P. tanacetifolia*), and the remaining species, joining a common cluster without further differentiation. 

In the case of clustering based on PCs from the rhizobox imaging traits (Figure 2E), legumes (here including *T. alexandrinum*) joined a common group (sp*R*^2^ = 0.017), having similar scores in all PCs. A second closely related cluster was formed by species distinguished mainly by their high PC2 scores (high visible length, narrow lateral branching distance) composed of the two brassicaceae as well as *P. tanacetifolia* and *H. annuus*, whereas *F. esculentum* joined this group at a distance of sp*R*^2^ = 0.082. *L. usitatissimum* and *A. strigosa* were allocated separately (joining at a sp*R*^2^ = 0.242) with similarity mainly in PC1 scores (primary axis dominated system with low convex hull), whereas contrasting in PC3 scores (fineness of axes).

Integrating lower-level root descriptors from the two data sources (rhizobox + Root Atlas) again allocated legumes in a common cluster, closely joined by *A. strigosa* (separation at sp*R*^2^ = 0.006) with similar scores on all PCs. The second closely related group contained the two brassicaceae (*R. sativus* and *C. sativa*; sp*R*^2^ = 0.070) with similar scores in the latent variables capturing branching pattern (PC1) and distribution (PC2). Linkage of the brassica-cluster with *P. tanacetifolia* occurred at a sp*R*^2^ of 0.113. *H. annuus* and *F. esculentum* were allocated to a common cluster at a sp*R*^2^ = 0.098, with high scores in PC1 and negative PC2 scores (and differing in PC3). Again, *L. usitatissimum* was disjoined from the other species, separating at a sp*R*^2^ = 0.165 from the nearest (legume/*A.strigosa*) group.

## 3. Discussion

### 3.1. Root Data Integration

Root datasets are quickly increasing with advance in measurement methods in the context of plant phenotyping (e.g., 1.2 million root images for *Arabidopsis thaliana* and 58,000 for rice; [27]). Assuming that each method provides a snapshot of the root system under specific environmental conditions and phenological stage(s) dictated by the experiment, integration of root data, even across distant methods, could provide a more biologically comprehensive view. Following this idea, we investigated the joint usability of data from image-based root phenotyping [28] and field descriptions of the Root Atlas series [29,30] for a data-driven classification of cover crop root systems [31]. 

We extracted two types of quantitative information from Root Atlas: (i) botanical expert knowledge from the text (maximum depth, highest order), and (ii) single plant data (total length) from the hand-drawn in-situ root excavations. Our results demonstrated that standardized rhizobox root imaging data had significant correlations with quantitative Root Atlas descriptors. The closest cross-method linkage was found between Root Atlas branching order and imaging-based single axis branching descriptors (Figure 1; *R* = 0.75, *p* = 0.006 for convex hull and *R* = 0.83, *p* = 0.019 for primary-lateral ratio). Maximum rooting depth from the Root Atlas was significantly related to convex hull (*R* = 0.70, *p* = 0.0344) and rhizobox-measured total length relation (*R* = 0.68, *p* = 0.0445). Correlation results thus suggest that, in particular, the generalized quantitative information from the text provides a robust species descriptors. 

Root length, derived from hand-drawn excavations, on the contrary did not show significant correlation to the imaging data, which suggests high environmental specificity and reveals the inference problem towards species characterization from this type of non-replicated botanical field description. Better results might be expected for species excavated at several locations (e.g., maize), thereby reducing the influence of single plant–environment interaction. However, also other results on root length from replicated measurement, providing a better representation of the expected population average, demonstrate that root length is a strongly adaptive trait, which does not easily scale across observation systems. For example, comparison of wheat seedling root length between different experimental systems (gel, sand, field) showed changing ranks among varieties [32]. Particularly for monocotyledons with a dominant shoot-borne root system, inter-method comparison of root length is also strongly dependent on phenological stage [33]. 

In this study, root trait relations across observational methods were investigated via linear correlation only. Therefore, potential non-linear patterns linking simple and complex traits, phenological stages, or environmental conditions were not considered. For root length and depth distribution, application of root architecture models could essentially improve integration of different experimental systems by biologically meaningful scaling across phenological stages and environments [34]. With the relevant environmental (weather, soil hydraulic properties, and nutrient status) and phenological metadata (flowering date), and a minimum degree of taxonomic standardization (e.g., [4,35]), model-based scaling could substantially advance root data integration and provide important added value to root system understanding and species/variety description.

### 3.2. Root System Descriptors for Comprehensive Species Characterization

Beyond the challenge of data integration, application of different experimental setups to the same species/cultivar provides varying descriptors at different scale for the same biological organism and their expression under diverse environmental conditions. Correlation results revealed lower-level traits related to more complex higher-level descriptors, as well as independent descriptors not expressed by any others. 

Within the single axis descriptors, the angle of lateral root emergence from the primary axis did not correlate with any other trait and did not scale to deep rooting (top-to-bottom length ratio). Lateral root growth direction depends on the insertion angle of initially agravitropic laterals [36] and their subsequent bending driven by several environmental cues (gravity, water, nutrient, temperature gradients, mechanical resistance [37,38]). The root angle parameter in this study, measured by RootNav [39], captured the initial emergence angle, which significantly differed among the investigated species. However, its role in higher-level architecture both at single axis (convex hull) and whole system (top-to-bottom ratio) scale was obviously less important. The two species with highest lateral angle of emergence in this study (*L. usitatissimum*, *A. strigosa*) had very short laterals (ratio of primary-to-lateral length 2.8). As laterals start to bend only at a certain distance from their initiation (grativtropic set point [36]), this indicates that emergence angle might scale poorly to root depth distribution. Other studies showed the relation of convex hull with depth distribution (and lateral extension [40]). The relationship of single-axis hull with top-to-bottom ratio suggests that this parameter provides an integrative descriptor for emergence angle and lateral bending/tropism that better scales to the whole-root system than angle alone.

On the contrary, the distance between emerging laterals showed significant relations to several other traits, both at the single axis scale (convex hull, primary-to-lateral length ratio) as well as at the whole root system scale (total length, top-to-bottom ratio). Distance between emerging laterals has a genetic component of formation of lateral primordia [41] and an adaptive side controlling lateral emergence in response to environmental cues [42]. Inter-branch distance was found as a strongly distinctive among a set of 45 dicot species, with values in a similar range as measured here [43]. Enhanced root proliferation via narrow lateral-spacing has been targeted to optimize nutrient foraging potential of plants [44]. Inter-branch distance between lateral roots is thus highly relevant for species and cultivar description, both as a strong driver of the overall root architecture as well as a functionally relevant trait. However, its acquisition via root imaging systems using soil-grown plants is still challenging due to incomplete root visibility. (Semi)automatic extraction of root axes segments with minimum length for representative determination of inter-lateral distance would be an important advance for (high-throughput) phenotyping compared to the manual tracking used in this study. 

At the root system scale, an uncorrelated trait was fine root ratio/average root diameter (with close mutual relation). Different parameters of fine rooting have been used in the context of root assimilate investment strategies and root functionality [45,46]. On the basis of a larger dataset, it was reported that root diameter was linked to the branching distance of laterals [43]. Also, relations of diameter with maximum root order might be expected [47]. We assume that variability in our relatively small dataset (10 species; CV_diameter_ = 0.23; CV_fine rooting_ = 0.34) was insufficient to discover more subtle relations with other parameters such as inter-branch distance. However, measurement accuracy of diameter/fine root proportion from rhizobox images is also critical—for example, we noticed that image resolution for species with intense root hair formation might have been insufficient, possibly overestimating diameter of root axes segments with root hairs (Figure 3).

Overall, we suggest a minimum-dataset for classification of mature root systems to include (i) interbranch distance, lateral angle, and maximum branching order (single axis descriptors); (ii) maximum rooting depth (as a proxy for root elongation rate not available in this study; upscaling descriptor to whole root system size); and (iii) fine root proportion (assimilate investment descriptor). These parameters and metadata on phenology (flowering date) and environmental conditions would also allow running mathematical models such as CRootBox [48] for functional assessment of distinctive root system.

### 3.3. Cover Crop Rooting Types

Cover crop species were classified according to similar rooting characteristics by a data-driven clustering approach [31]. Three sets of traits were used for PCA and subsequent clustering (Root Atlas data only, rhizobox data only, integration of rhizobox and Root Atlas data with variables selected according to minimum-dataset requirements discussed in Section 3.2) to compare trait-dependence of the resulting clusters. 

Trait allocation towards principal components suggested composite root variables for classification capturing (i) branching pattern, (ii) size and depth distribution, and (iii) fineness (assimilate investment; for rhizobox and integrated dataset). Independent of the input data, legumes were allocated into a common cluster, suggesting strong distinctiveness of their rooting pattern. Extending root system characterization beyond the limited set of Root Atlas descriptors with morphological and architectural imaging traits resulted in more detailed species distinction. Brassicaceae (*R. sativus*, *C. sativa*) conserved a common cluster, with higher distinctiveness from the other size-dominated species (high length, narrow lateral branching; *P. tanacetifolia* and *H. annuus*) when integrating imaging with Root Atlas data (expressed by the more distant linkage to *P. tanacetifolia* and separation of *H. annuus* towards a different cluster). *L. usitatissimum* tended towards a separate cluster, although still in close linkage to the legume/oat group in case of using Root Atlas descriptors only. In spite of similar scores of *L. usitatissimum* compared to other primary-axis dominated species such as *A. strigosa*, *V. faba*, and *V. sativa* in terms of branching and size-related principal components, imaging data revealed its strong distinctive via fine root proportion. *F. esculentum* and *H. annuus* had the most variable cluster allocation, depending on the dataset. *F. esculentum* was allocated in close vicinity to *P. tanacetifolia* with Root Atlas data only, and rhizobox imaging traits suggested closest linkage to the size-dominated rooting cluster (brassicaceae + *P. tanacetifolia;* sp*R*^2^ = 0.082). For the integrated dataset, *F. esculentum* was closest to *H. annuus* at a sp*R*^2^ = 0.098, whereas in this case the latter was disjoined from the size-related brassicaceae + *P. tanacetifolia* group due to its different allocation at PC2 (distributional pattern).

Overall, the classification results suggest a weak plant family-related delineation among root system types with large inter-family overlaps even in case of larger datasets. This suggests following Fitter’s [6] approach of understanding distinctive rooting types mainly from their functional implications. In this sense, for the cover crop sample investigated here, we suggest the following functional groups: (i) legume group (*V. sativa, V. faba, T. alexandrinum*), (ii) root density group (closest similarity among *R. sativus*, *C. sativa, P. tanacetifolia;* dataset dependent linkage to a subgroup of *H. annuus* and *F. esculentum*), and (iii) lower-order axes dominated species (*L. usitatissimum*) with dataset-dependent weak linkage to *A. strigosa*. 

Legume group: A common feature of the legume rooting type was its low lateral emergence angle (65.4°; significant contrast to all others). At the same time, the legume group showed the lowest root length allocation to deep layers (7.9%; significantly different from dense rooting group only). Legumes were intermediate in convex hull area (integrating angle and bending, see discussion in Section 3.2), significantly influencing top-to-bottom ratio/maximum depth (cf. Figure 1). Functionally, root allocation has been extensively studied in relation to nutrient foraging, with large variability even at the genotypes level due to difference in basal root insertion angle and gravitropic response (e.g., *Phaseolus vulgaris*; [49]). The preferential topsoil allocation of root axes in the autumn-grown temperate cover crop legumes, in spite of their steeper lateral emergence angle, could indicate their predominant adaptation to non-mobile nutrient foraging (in the case of legumes, particularly phosphorus) over deep-soil water and nitrate exploration. Morphologically, a high contribution of larger diameter root axes on total root length has been shown for legumes (e.g., low specific root length [50]). This was the case for *V. sativa* and *V. faba*, but not for *T. alexandrinum*. At the single axis-scale, legumes shared an intermediate position for the ratio of (larger diameter) primary-to- (finer) lateral length ratio (*p_contrast_* = 0.012 and 0.006), which, however, did not scale significantly to the whole root system proportion of fine root length. *A. strigosa*, joining the legume group with the reduced dataset, is mainly related to a similarly large root diameter/low fine root ratio. Similarity is also found in low convex hull and lateral length contribution, which equally characterizes *L. usitatissimum*, being, however, fine rooted (i.e., contrasting in PC3). Agro-environmental benefits of surface-allocated larger diameter legume root systems have been discussed mainly in relation to physical effects on macroporosity [51], as well as their particular role in biological interactions (rhizodeposition, high-N residues) with soil microbes [52,53,54]. 

Low-order axes types: *L. usitatissimum* and *A. strigosa* are both distinguished by their primary axes dominance ratio with low contribution of laterals (maximum third order according to Root Atlas) that emerge in a near-horizontal angle, resulting in the smallest convex hull. A sparsely branched (herringbone) root topology was previously described as a key distinctive pattern of Poaceae compared to other (more dichotomously branched) plant families [55]. The reason for the still weak link of *A. strigosa* to *L. usitatissimum* is due to differences in fine rooting (contrasting PC3). In case of *A. strigosa* (in tillering stage when imaged) this can be probably explained by a stronger assimilate sink of elongating (large diameter) shoot-born roots over finer laterals. *L. usitatissimum* was among the species with highest inter-branch distance (similar to *V. sativa* and *V. faba*), with soil exploration largely depending on low-order axes (according to Root Atlas data, maximum third order laterals). Interestingly, highest mycorrhizal root colonization in *L. usitatissmum* was described among a sample of eight cover crop species [56]. Other studies also reported high arbuscular mycorrhiza fungi colonization rate of *L. usitatissmum* [57]. This observation suggests that the sparsely branched rooting type of *L. usitatissimum* is related to strong mycorrhizal dependence, in accordance with suggested root trait-mycorrhiza linkages [58]. Functionally grass cover crops (oat, rye, ryegrass) have been frequently found to be particularly beneficial for aggregate stability and soil organic carbon [59,60,61], related to both root enmeshment [62] and excretion of organic binding agents into the soil [59]. Secondary shoot-born axes are well known for their formation of rhizosheath, revealing a close root–soil interaction [63]. To our knowledge, there is no investigation on soil influence of *L. usitatissmum*. It could be speculated that the high mycorrhizal colonization, as well as the chemical recalcitrance of its root (and shoot) bast fibers, would also provide benefits for soil structure and organic carbon reproduction.

Root density group (*R. sativus, C. sativa, P. tanacetifolia*; with weaker linkage to *F. esculentum*, forming a trait dependent group with *H. annuus*): Linear contrasts reveal the significant differences in total root length, mediated by lowest distance between laterals and highest convex hull. Separation of *F. esculentum/H. annuus* with the reduced dataset was due to a negative PC2 (positive for the other three species), suggesting a different distribution pattern (in spite of similarity in ANOVA for the single variables angle and top-to-bottom ratio). Among the density rooting types, species with visually observed highest root hair formation were also found (cf. Figure 3). In particular, the rooting type of *F. esculentum*, with more topsoil-allocated high proportion of fine roots (Table 1), highest branching order (Table 3), and intense root hair formation, suggests strong soil foraging capacity, which was demonstrated for P (also related to soil acidification [64,65]). Some studies also indicate *P. tanacetifolia* (in our study—highest convex hull, high fine root proportion, strong root hair formation) as a potential P-mobilizer [66]. *R. sativus*, *C. sativa*, and *H. annuus* had the highest root length, with the two Brassica species also showing highest density in lateral branch spacing. *R. sativus* had the significantly highest capacity for soil exploration by deep root allocation, followed by *H. annuus*. Species from the density group are therefore the best candidates for cover crops to mitigate nitrate leaching, as confirmed by other studies (e.g., [24]).

## 4. Materials and Methods 

### 4.1. Plant Material

A total of 10 plant species were selected (i) on the basis of their importance for the agro-environmental practice of post-harvest (July/August) cover cropping in temperate cropping systems, and (ii) covering a wide range of plant families (Table 4).

### 4.2. Rhizobox Experiment

#### 4.2.1. Experimental Setup and Growing Conditions

A rhizobox experiment was established in a greenhouse at the University of Natural Resources and Life Sciences, Vienna (BOKU) at Tulln to obtain root parameters by image analysis of the 10 cover crop species. Rhizoboxes (30 × 100 × 1 cm) consisting of a PVC (polyvinyl chloride) frame with a mineral glass observation window for measuring visible roots. During plant growth, rhizoboxes were held at a 45° angle by a metal frame to gravitropically maximize root visibility at the front glass. (For detailed description of the rhizobox setup, see [28]).

The soil used for the experiment was field topsoil (calcareous chernozem; sand 0.25 g g^−1^, silt 0.53 g g^−1^, clay 0.22 g g^−1^; pH 7.1; soil organic carbon: 1.1%; plant available water: 0.25 cm^3^ cm^−3^) sieved to 2 mm and filled at a bulk density of 1.1 g cm^−3^. Soil moisture during the experiment was 77.4 (± 3.4) % of plant available water. Irrigation water was amended with liquid NPK fertilizer (6% N, 3% P_2_O_5_, 6% K_2_O) to exclude nutrient limitation. Average maximum air temperature was 18.0 °C and minimum temperature was 11.4 °C.

Two seeds per species (replicate number per species *n* = 3) were placed in each rhizobox with an average distance of 12.5 cm between each other, representing the common plant spacing in field-sown cover crop stands. The experiment was terminated when roots approached the bottom of rhizoboxes between 21 days (*R. sativus, H. annuus, F. esculentum*) and 39 days (*L. usitatissimum*) after sowing. (We noticed that the experiment was continued after cutting the shoots to further investigate root decomposition; not reported here).

#### 4.2.2. Imaging and Image Analysis

RGB images of root systems with all types of root axes developed (dicots: primary, basal, and their respective laterals; *A. strigosa*: primary, basal, and shoot-born roots with their respective laterals; taxonomy following [4]) were taken via the front glass. Rhizoboxes were inserted into an imaging box to shield from ambient light. They were artificially illuminated by four 24 W fluorescent light tubes and images were captured with a Fuji X-Pro1 camera (16.3 Megapixels, resolution: 0.1 mm pixel^−1^). For details of image box construction and recommended camera settings, refer to [28]. 

Images were then analyzed for root morphological and architectural parameters. For root morphological analysis (total root length, root diameter, diameter distribution of root length) we used WinRhizo Pro upon color-based segmentation between roots and background. Root parameters were acquired in 10 cm increments to derive root depth distribution.

For root architecture parameters, we manually tracked a primary root segment in the lower 30 cm of the rhizoboxes, starting from the visible root tip. Selection of the apical 30 cm for evaluation of branching was due to strong root overlap in the top compartment of the rhizoboxes, which did not allow us to manually track individual axes. Root tracking was performed using a graphic tablet (Wacom Intuos Pro) in CorelDraw X7. The manually tracked root axis was then analyzed for (i) distance between laterals in the branching zone and the ratio of lateral-to-tap length using Root System Analyzer [67] and (ii) average branching angle of laterals emerging from the tap root and the convex hull area covered by the measured axes using RootNav [39].

Figure 4 provides an example of the analytical pipeline for the rhizobox experimental approach from the original RGB images to the analysis of visible root system traits and single axis branching pattern.

### 4.3. Root Atlas Data

Hand-drawn images of in situ excavated root systems of the different cover crop species were taken from the Root Atlas series (7th volume, [29]; except *V. sativa*, 1st volume, [28]). The drawings represent root systems of single plants (*n* = 1) at flowering stage under specific environmental conditions. Images were scanned in high resolution (1200 dpi) for measuring root length using WinRhizo. This measurement was complemented with quantitative information contained in the Root Atlas series on maximum rooting depth and highest order of branching (an example is given in Figure 5).

### 4.4. Statistical Evaluation

All species parameters from the rhizobox experiment were compared for a significant effect (*p* < 0.05) by analysis of variance using SAS PROC GLM. In case of significance, comparison of means was performed by a Tukey–Kramer post-hoc test. Differences of single parameters for groups of species (obtained from cluster analysis; see below) were compared by linear contrasts (CONTRAST statement in PROC GLM).

Correlations between parameters were obtained using the SAS procedure PROC CORR in order to determine (i) if standardized rhizobox root data are related to parameters from in situ field excavations, and (ii) which types of parameters are uncorrelated, and which traits are linked across different observation scales (single axis, total root system, field). Significance of correlations (H_0_ = no correlation) were tested with a chi-square goodness-of-fit test (*p* < 0.05).

Distinctive rooting types among the species were determined by multivariate statistics (principal component analysis (PCA) and clustering) for a data-driven exploration of rooting types [31]. Two parameter sets were used to compare the resulting grouping of species: (1) parameters from the rhizobox experiment only (morphology: total root length, proportion of roots below 70 cm soil depth, proportion of fine roots < 0.4 mm diameter, average root diameter; architecture: lateral distance, emergence angle, primary-to-lateral length ratio, and convex hull area); and (2) on the basis of the results from correlation analysis, lower-level traits scaling across observation systems and uncorrelated traits from both data sources (rhizobox, Root Atlas), that is, lateral distance, lateral angle, and maximum branching order; maximum rooting depth and fine root proportion were used. PCA was performed using SAS procedure PROC FACTOR and clustering was done using PROC CLUSTER with Ward’s minimum-variance method. The dendrogram was constructed with PROC TREE.

## Figures and Tables

**Figure 1 plants-08-00514-f001:**
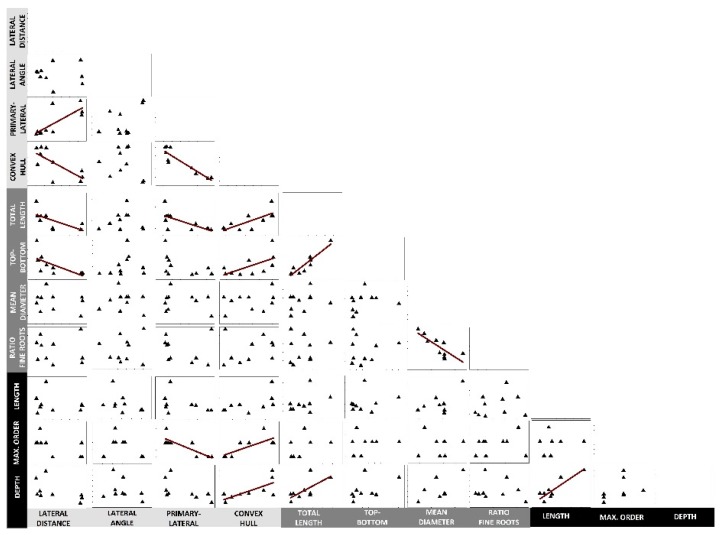
Inter-trait relations of cover crop root systems. Parameters with light grey background are architectural branching descriptors from a single primary axis of rhizobox plants, parameters with dark grey background are morphological parameters from all visible roots of rhizobox grown plants, and parameters with black background are descriptors obtained from Root Atlas series. Dark red linear trend lines show significant (*p* < 0.05) relations between parameters.

**Figure 2 plants-08-00514-f002:**
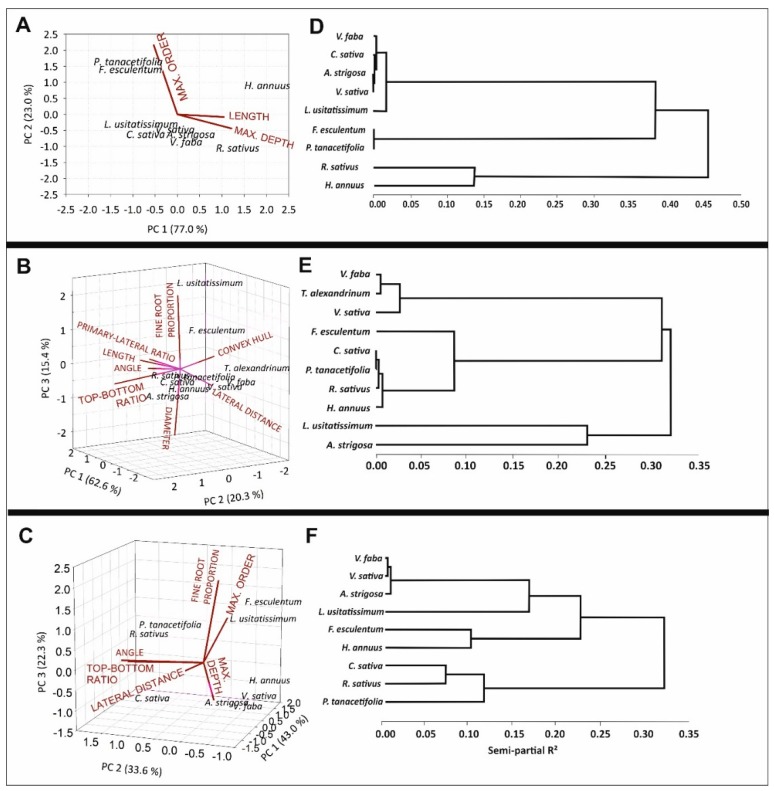
Biplots from principal component analysis (left; **A**–**C**) and dendrograms from cluster analysis (right; **D**–**F**) for cover crops using (**A**,**D**) Root Atlas data only (*T. alexandrinum* not described in Root Atlas), (**B**,**E**) morphological and architectural root traits obtained from rhizobox imaging, and (**C**,**F**) lower-level root traits integrating rhizobox imaging and Root Atlas data. The biplots visualize trait vectors (red lines; similar vector direction indicates correlation among traits; vector length indicates trait importance for building the principal components) and location of genotypes according to principal component scores. See Table 1, Table 2 and Table 3 for trait description. Dendrograms cluster the cover crop species into distinctive groups based on their root principal component scores. Clusters with small semi-partial *R*^2^ show high homogeneity in their rooting pattern.

**Figure 3 plants-08-00514-f003:**
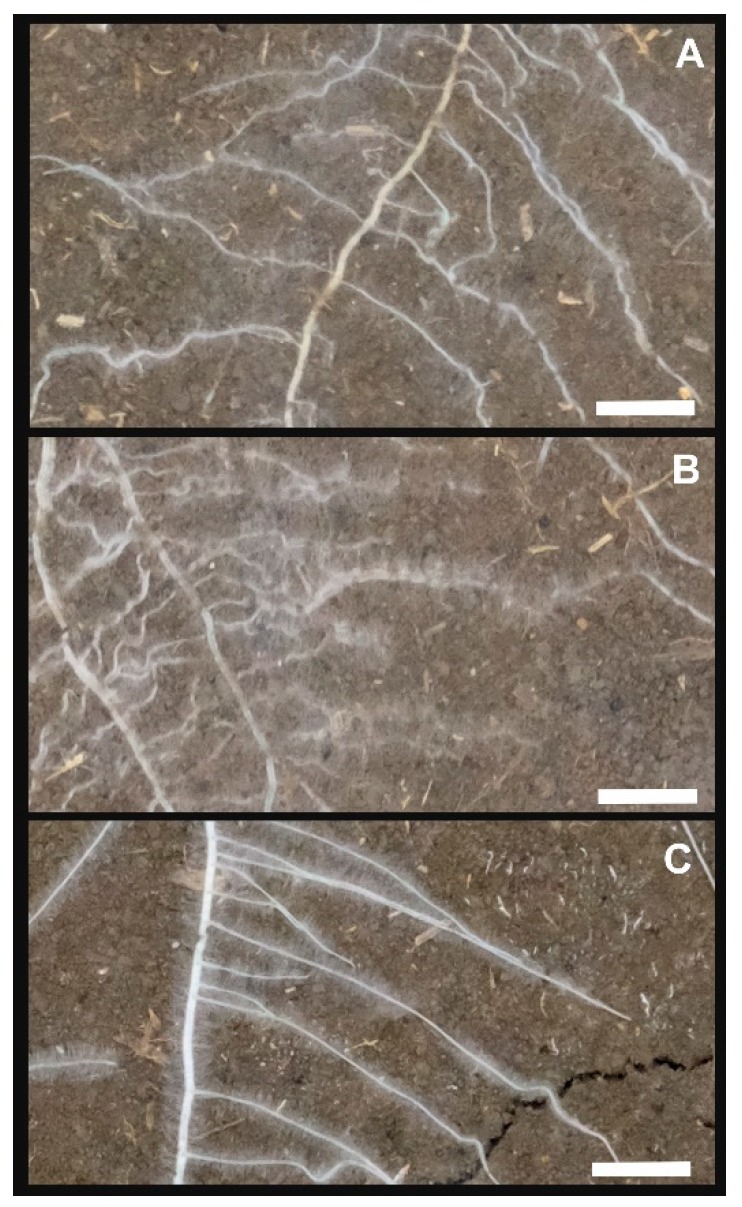
Species with strong root hair formation. (**A**) *F. esculentum*, (**B**) *P. tanacetifolia*, (**C**) *H. annuus*. White bar at the bottom right side of the images is 1.0 cm.

**Figure 4 plants-08-00514-f004:**
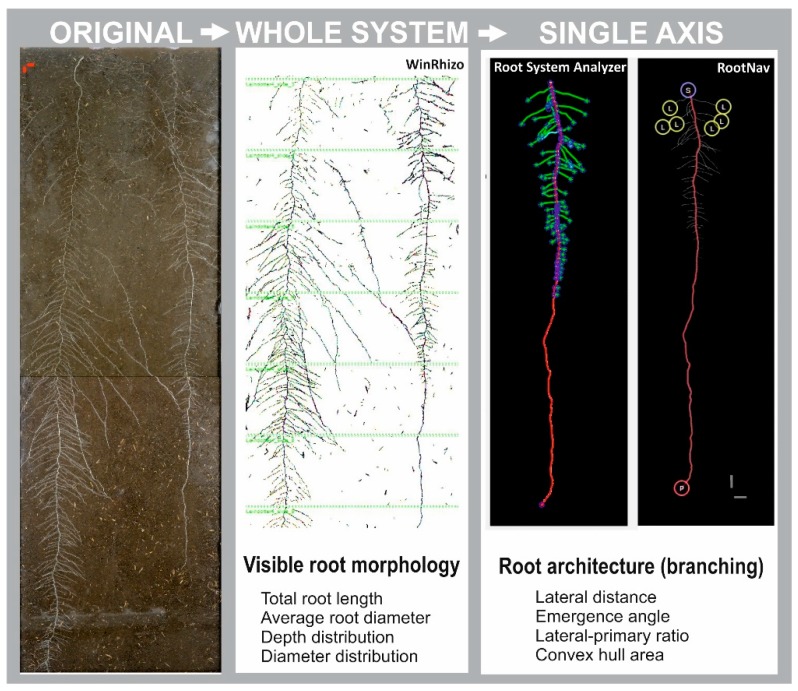
Image analysis procedure. Left: original RGB image; middle: color-segmented image in WinRhizo subdivided into 10 cm depth segments (light-green lines) for measurement of root morphological parameters; right: manually tracked single axis in Root System Analyzer and RootNav for measuring architectural (branching) traits.

**Figure 5 plants-08-00514-f005:**
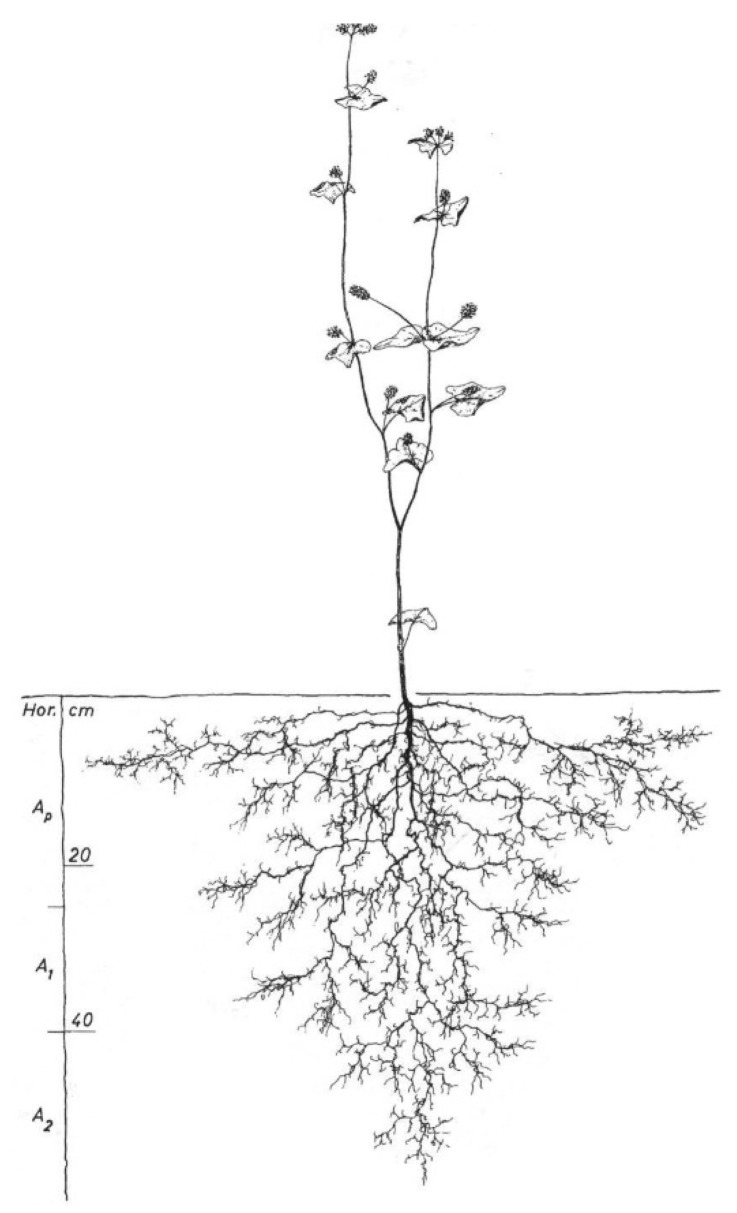
Example of field excavated, hand-drawn root system from the Root Atlas series. *Fagopyrum esculentum*; height_shoot_: 82 cm, depth_root_: 58 cm, lateral extention_root_: 75 cm; location: St. Donat, Carinthia, 482 m NN., excavation: 23 June 2003. Soil type: colluvial Cambisol [29].

**Table 1 plants-08-00514-t001:** Morphological characteristics obtained from visible roots at the surface of rhizoboxes.

Species	Total Length (cm)	Top–Bottom Ratio ^†^ (relative)	Diameter (mm)	Fine Root Ratio ^††^ (relative)
*Vicia faba*	570.5 f	0.051 c	1.02 b	0.278 d
*Vicia sativa*	2075.3 bcd	0.070 bc	0.94 bc	0.336 d
*Trifolium alexandrinum*	743.4 ef	0.076 bc	0.73 cde	0.498 bc
*Camelina sativa*	2694.9 b	0.138 bc	1.02 b	0.315 d
*Raphanus sativus*	6164.8 a	0.265 a	0.88 bcd	0.481 c
*Fagopyrum esculentum*	1498.1 cde	0.089 bc	0.68 de	0.610 ab
*Phacelia tanacetifolia*	2245.4 bc	0.126 bc	1.01 b	0.513 bc
*Helianthus annuus*	2661.7 b	0.151 b	1.33 a	0.331 d
*Avena strigosa*	1274.8 def	0.138 bc	1.02 b	0.251 d
*Linum usitatissimum*	1008.0 ef	0.075 bc	0.58 e	0.673 a
*p*-value	<0.001	0.005	<0.001	<0.001

**^†^** Proportion of visible root length in 70–100 cm relative to total length; **^††^** ratio of root length with diameter < 0.4 mm to total length. Species sharing common lower-case letters for a given trait are not significantly different at *p* < 0.05.

**Table 2 plants-08-00514-t002:** Architectural characteristics of branching pattern obtained from a single (30 cm) primary root axis (manually traced).

Species	Lateral Distance (cm)	Lateral Angle ^†^ (degree)	Primary-Lateral Ratio ^††^ (rel.)	Convex Hull (cm^2^)
*V. faba*	1.3 a	71.7 bc	1.67 a	102.0 abc
*V. sativa*	1.3 a	65.5 cd	1.91 a	76.3 bc
*T. alexandrinum*	0.6 b	59.1 d	0.28 b	138.9 abc
*C. sativa*	0.2 de	75.3 b	0.31 b	128.9 abc
*R. sativus*	0.2 e	75.5 b	0.11 b	199.3 a
*F. esculentum*	0.3 cde	71.4 bc	0.16 b	173.5 ab
*P. tanacetifolia*	0.3 cd	76.4 b	0.20 b	202.9 a
*H. annuus*	0.4 bc	70.5 bc	0.43 b	199.8 a
*A. strigosa*	0.6 b	84.8 a	2.60 a	53.1 c
*L. usitatissimum*	1.3 a	85.3 a	2.81 a	57.7 c
*p*-value	<0.001	<0.001	<0.001	0.039

**^†^** Angle of emergence of first order laterals from primary axis; **^††^** ratio of primary root length to length of laterals. Species sharing common lower-case letters for a given trait are not significantly different at *p* < 0.05.

**Table 3 plants-08-00514-t003:** Quantitative root characteristics from Root Atlas series.

Species	Length (cm)	Depth (cm)	Highest Order (Number)
*V. faba*	2053.2	90	4
*V. sativa*	1933.1	80	3
*T. alexandrinum*	-	-	-
*C. sativa*	841.1	93	4
*R. sativus*	2946.6	153	4
*F. esculentum*	1730.7	105	5
*P. tanacetifolia*	2061.6	88	4
*H. annuus*	5291.0	180	4
*A. strigosa* *	1265.6	92	3
*L. usitatissimum*	1259.6	60	3

* Characteristics taken from *Avena sativa*.

**Table 4 plants-08-00514-t004:** Cover crop species (species name and family) characterized by rhizobox experiments and Root Atlas data.

Species	Family
*Vicia faba* L. (Faba bean)	Leguminoseae
*Vicia sativa* L. (Common vetch)	Leguminoseae
*Trifolium alexandrinum* L. (Berseem clover) ^1^	Leguminoseae
*Camelina sativa* (L.) Crantz (Cemelina)	Brassicaceae
*Raphanus raphanistrum* subsp. *sativus* (L.) Domin (Radish)	Brassicaceae
*Fagopyrum esculentum* Moench (Buckwheat)	Polygonaceae
*Phacelia tanacetifolia* Benth. (Phacelia)	Boraginaceae
*Helianthus annuus* L. (Sunflower)	Asteraceae
*Avena strigosa* Schreb. (Black oat) ^1^	Poaceae
*Linum usitatissimum* L. (Linseed)	Linaceae

^1^ Not described in Root Atlas series. For *A. strigose*, Root Atlas data of *A. sativa* is used.

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
