# Peer review of "Characterization of Cover Crop Rooting Types from Integration of Rhizobox Imaging and Root Atlas Information"

_plants, 2019, doi:10.3390/plants8110514_

Round 1

Reviewer 1 Report

Address the following points in the Materials and Methods section:

Although studying 10 cover crop species is good, data collected from two seedlings per species hardly provide strong evidence to achieve the objectives. For example, in line 62, “root-environment interactions are too complex”. One can also point out that “species-root-environment” are too complex. Therefore data collected from 20 seedlings (10 species x 2 seedlings each) cultured in the same soil under same growing conditions did not provide much value for the objectives. Imaging taken when the taproot reached the bottom of the rhizoboxes was called “final imaging” (line 389). Were there any other times during the study that imaging was taken? If so, how often? There were no data documented on imaging taken other than the final imaging. Define “fully developed root systems” in line 394 as opposed the fully developed root system (line 419) shown in Fig 7 Were the two root systems in Figure 6A and 6B from of the same species? At what point was lateral root diameters measured? Was the value of average diameter from taproots and all lateral roots? Diameter distribution of root length was not reported. Why was root architecture parameters only assessed from the lower 30 cm of the rhizoboxes? Were the measured parameters of similar values as the upper 70 cm? Plant shown in Fig 7 from field excavation had 58 cm root depth. Was the single axis (Table 2) always the taproot?Results and Discussions   Line 98: C. sativa, H. annuus, and V. sative also had similar fine root lengths compared to V. faba and A. strigosa. Table 2: align p-value Make sure names of all root parameter descriptors are consistent throughout the manuscript. For example, top-bottom ratio (lines 142, 148, and 170; Fig 1) vs deep rooting (Table 1). T. alexandrinum was misspelled in several places. Fig 2A: missing genus name for tanacetifolia. Fig 4A: need higher resolution for this picture. Line 336: spell out AMF. Figure 5 can be deleted. A short description of the growing conditions should suffice.

Author Response

Note: all line numbers refer to the revised manuscript.

Address the following points in the Materials and Methods section:

Although studying 10 cover crop species is good, data collected from two seedlings per species hardly provide strong evidence to achieve the objectives. For example, in line 62, “root-environment interactions are too complex”. One can also point out that “species-root-environment” are too complex. Therefore data collected from 20 seedlings (10 species x 2 seedlings each) cultured in the same soil under same growing conditions did not provide much value for the objectives.

A: Here the reviewer addresses a common root phenotyping challenge: is the inference from a defined controlled phenotyping setup towards a root system growing under natural environmental conditions within a crop stand valid. In case of (image based) root phenotyping, measurements are almost exclusively based on single plants to capture their detailed root architecture not accessible in field samples (generally restricted to morphology). Furthermore, phenotyping pretends to infer from a limited set of pre-defined treatments like “optimum conditions” (as in this manuscript) or a (single) stressor  - with the non-trivial challenge of treatment levels particularly in case of mobile resources - to highly variable spatio-temporal patterns in natural environments.

Our main aim is to present/exemplify a concept and method to face this challenge via data integration among different observational systems for better capturing distinctive root system types within their natural plasticity - that is represented in the variable settings of experiments.   

The rhizobox phenotyping experiment was set up for “optimum conditions”, i.e. not targeting interacting effects of roots with environmental stresses. For this reason, we kept water and nutrients at optimum (near to 80 % plant available water and regular addition of liquid NPK-fertilizer; information added in line 433), assuming that the observed differences largely represent constitutive expressions of the species-specific rooting pattern.

The reference to (species-specific) root-environmental interactions in line 63 refers to the general challenge of representing the dynamic and multiple stresses in field environments within a single phenotyping experiment. With this reference we intended to introduce the need for unifying datasets from different experiments (=environments), establishing root databases with the respective metadata documentation on stress levels, and developing methods to merge the obtained descriptors towards a comprehensive classification of species (genotypes) – either via multivariate statistical identification of groups of similar species (genotypes) or also via mathematical models (see discussion line 316-318). In the case of this study, our intention is to exemplify this concept by merging the image-based phenotyping data with the (methodologically strongly different) descriptions from the Root Atlas series.

To better clarify data integration and the subsequent species classification (also following a suggestion of Reviewer 3) we now show/discuss in more detail the impact of grouping (classification) from the two data sources separate (rhizobox, Atlas) and the grouping resulting from the merged-data classification (see revised Figure 2 and discussion lines 325 ff).

In addition, we reformulated the respective part in the Introduction to avoid claiming that the results presented here pretend to assess “species-root-environment interaction”, but mainly focuses on exemplifying the potential of data integration to approach such a description (lines 66 ff and 86-88).       

Imaging taken when the taproot reached the bottom of the rhizoboxes was called “final imaging” (line 389). Were there any other times during the study that imaging was taken? If so, how often? There were no data documented on imaging taken other than the final imaging.

A: Here we only report a single time-point imaging based on two criteria: root axes elongated to the bottom of the rhizobox (1 m) and all root axes types present (see next comment on “fully developed”). We deleted the word “final imaging” and clarified the criterion for finishing the experiment (line 437 ff).

Define “fully developed root systems” in line 394 as opposed the fully developed root system (line 419) shown in Fig 7

A: We corrected the misleading formulation, specifying that in case of rhizoboxes we refer to a root system with all axes types present (for monocots including shoot-borne root with their laterals), and in case of the Atlas Series plants are measured/documented at flowering (line 471; which is often [not always] reported as phenological stage with maximum development of roots due to changing sink towards reproductive organs).

Were the two root systems in Figure 6A and 6B from of the same species?

A: Yes, Fig 6A and 6B are the same species (in the same rhizobox).

At what point was lateral root diameters measured? Was the value of average diameter from taproots and all lateral roots? Diameter distribution of root length was not reported.

A:Average diameter was measured from the images when roots have elongated to the bottom of the rhizobox (see above), with the value being the average of all visible roots (i.e. a weight average over [tap and lateral] length as measured in WinRhizo). We have now included the diameter distribution as a supplemental figure. However, notice (line 305 f) that generally the precision of diameter measurements from rhizobox images has to be taken with care due to resolution constraints.

Why was root architecture parameters only assessed from the lower 30 cm of the rhizoboxes? Were the measured parameters of similar values as the upper 70 cm?

A: Assessing root architecture in mature root systems grown in rhizoboxes, particularly with two individuals and the resulting stronger overlap between axes, is hardly possible with sufficient precision and for most parameters. Therefore, we used the apical part of the tap root and an axes length of 30 cm, i.e. covering the unbranched apical zone as well as the branching zone to quantify interbranch distances and lateral length. Reports on changes of branching pattern along the root axes are variable. Some authors state that inter-branch distances are constant (at least at the level of primordia; e.g. Dubrovsky and Forde, 2012, The Plant Cell 24, 4-14), others described changing inter-branch distances between basal and apical parts (e.g. Ito et al., 2006, Physiologia Plantarum 127, 260–267). The main reason of using the bottom third, however, was that (even the manual) individual axis-tracking of the top part was not feasible. We added an explanatory sentence in line 453.

Plant shown in Fig 7 from field excavation had 58 cm root depth. Was the single axis (Table 2) always the taproot?

A: The 58 cm depth refers to the individual shown in Fig. 7 (revised version Fig. 6). The data on “maximum rooting depth” for the respective species given in Table 2 is taken from Root Atlas species description, with is based on the overall observations from the Atlas authors (“expert knowledge”) beyond the individual excavation we showed in Fig. 6 as an example. The axis type for maximum depth in Root Atlas is not specified, mostly it will refer to the taproot (except for monocots), but it is not excluded that in field environments (like Root Atlas observations) single lateral roots penetrate the soil deeper than the taproot.

Results and Discussions  

Line 98: C. sativa, H. annuus, and V. sative also had similar fine root lengths compared to V. faba and A. strigosa.

Corrected (line 110)

Table 2: align p-value

Done.

Make sure names of all root parameter descriptors are consistent throughout the manuscript. For example, top-bottom ratio (lines 142, 148, and 170; Fig 1) vs deep rooting (Table 1).

Corrected (changed in Table 1 and Figure 2 and checked for the manuscript)

alexandrinum was misspelled in several places.

Checked and corrected.

Fig 2A: missing genus name for tanacetifolia.

The figure was changed (following a recommendation of reviewer 3; the P. for Phacelia is slightly hidded because of its position near to R. sativus.

Fig 4A: need higher resolution for this picture.

A: We maximized the image resolution. However, consider that the figure is a close-up from a RGB image with an 8 mm glass window in front of the object, which slightly compromises the visibility of very fine structures. Image resolution is now added in line 442.

Line 336: spell out AMF.

Corrected.

Figure 5 can be deleted. A short description of the growing conditions should suffice.

Deleted

Reviewer 2 Report

The manuscript entitled 'Characterization of cover crop rooting types from integration of rhizobox imaging and Root Atlas information' by Bodner and colleagues presents how field measured root descriptors from the classical Root Atlas series with traits from controlled-environment root imaging for ten cover crop species could be used to detect descriptors scaling between distant experimental methods. This provided traits for species classification, and its implications to cover crop ecosystem functions. This would be an interesting paper for those in root biology field, in agriculture and researchers interested how root systems can be used in different conditions.

Here are some suggestions:

page 1, line 32 - 'cover crops and Root Atlas' are already in the title and keywords could be removed

Page1, line 39 - E.g. change to e.g. 

Page1, line 40 -  change '(monocotyledons;' to (monocotyledons)

Page 2, line 48 - remove 'Also' at the beginning of sentence

Page 2- results section 2.1 - authors introduce for the first time plant species names used in this study and it is not until the reader gets to the method section, one gets to know the full species name. Please check if one requires to write in full species name when first introduced in the manuscript in this section

Page 3, line 104 - 'C. sativa' not italicized

Page 4, line 121- 'H. annuus' not italicized; line 122 - 'C. sativa' not italicized; line 123 to 124 - 'F. esculentum' and  'V. sativa' not italicized

Page 13, line 425-426 - 'Fagopyrum esculentum' not italicized

Please make sure that in the reference section that authors check the Reference List and Citations Style Guide for MDPI Journals as some titles in the  reference list are in capital letters. Please verify if species names are italicized as well.

Author Response

Note: all line numbers refer to the revised manuscript.

Comments and Suggestions for Authors

The manuscript entitled 'Characterization of cover crop rooting types from integration of rhizobox imaging and Root Atlas information' by Bodner and colleagues presents how field measured root descriptors from the classical Root Atlas series with traits from controlled-environment root imaging for ten cover crop species could be used to detect descriptors scaling between distant experimental methods. This provided traits for species classification, and its implications to cover crop ecosystem functions. This would be an interesting paper for those in root biology field, in agriculture and researchers interested how root systems can be used in different conditions.

Here are some suggestions:

page 1, line 32 - 'cover crops and Root Atlas' are already in the title and keywords could be removed

Keywords are changed now.

Page1, line 39 - E.g. change to e.g.

Corrected.

Page1, line 40 -  change '(monocotyledons;' to (monocotyledons)

Corrected.

Page 2, line 48 - remove 'Also' at the beginning of sentence

Corrected.

Page 2- results section 2.1 - authors introduce for the first time plant species names used in this study and it is not until the reader gets to the method section, one gets to know the full species name. Please check if one requires to write in full species name when first introduced in the manuscript in this section

According to the Journal style, Material and Methods follows the discussion section. Therefore, following the note of the reviewer, we spelled the full names in the Results section once they first appear.

Page 3, line 104 - 'C. sativa' not italicized

Corrected

Page 4, line 121- 'H. annuus' not italicized; line 122 - 'C. sativa' not italicized; line 123 to 124 - 'F. esculentum' and  'V. sativa' not italicized

Corrected

Page 13, line 425-426 - 'Fagopyrum esculentum' not italicized

 Corrected

Please make sure that in the reference section that authors check the Reference List and Citations Style Guide for MDPI Journals as some titles in the  reference list are in capital letters. Please verify if species names are italicized as well.

References and species name layout have been checked.

Reviewer 3 Report

This paper describes the classification of cover crop roots systems from rhizobox experiments, using data from RootAtlas as a reference.  While the theory of the overall method has been described in a previous paper, this paper describes the multivariate method applied to a new dataset.  Initially this paper is quite descriptive.  The main issue I have is that the lack of detail on replicates, exactly how the clustering of the Rhizobox plus Root Talas was performed, make it almost impossible to evaluate the statistics properly.  Also what is the real aim here, is it to link the data to a standard reference point?

I would need to know the number of replicates and how what appears to be a single measurement from the root atlas was linked to what I assume was multiple replicates from the rhizobox experiment.  I would like to see a PC plot and dendogram of the Root Atlas data alone, to see how closely these relate to the rhizobox, or atleast the rhizobox positions data points marked separately from the Root Altlas data on the PC plots.  This leads onto the question of whether the clustering in the dendogram sample groups are grouped like that because the rhixobox data is driving the separation  (in Fig 3 b because the rhizobox is driving that data split.

Line 52 – “rely” rather than “relay”

Table 2, why is V.faba Underlined and in bold?

Table 2 Change the a and b reference links in the titles to symbols

Line 108 (CV 103.3 %)  CV should be given in a consistent format see Line88

Line 180 “joined” rather than “joint” (and Line 182)

Line 250 “angle” not “angel”

Did the authors check the likely hood of getting significant correlations by chance when doing the multiple correlations?

Line 387 – 392 – Number of replicates?

Line 397, please state resolution of images used for this study.

Line 417 – Section Root Atlas – So if I understand this correctly, for each of the root atlas species, there is effectively a sample of 1.

Why was the dataset of cover crops from ref 30 also not used in this study?

Author Response

Note: all line numbers refer to the revised manuscript.

This paper describes the classification of cover crop roots systems from rhizobox experiments, using data from RootAtlas as a reference.  While the theory of the overall method has been described in a previous paper, this paper describes the multivariate method applied to a new dataset.  Initially this paper is quite descriptive.

A: The key idea of this paper – beyond the descriptive part on a set of cover crops which, being minor crops utilized mainly in the context of agro-environmental programmes, largely lack a proper root description in literature (line 80) compared to other (cash) crops -  was the application of the multivariate approach we previously had applied to separate datasets from single experiments (i.e. data obtained by the same method) to datasets obtained with different methods, thereby integrating different descriptors for a species. We added a sentence (line 86-88) that clarifies this main idea of data integration.

The main issue I have is that the lack of detail on replicates, exactly how the clustering of the Rhizobox plus Root Talas was performed, make it almost impossible to evaluate the statistics properly.  Also what is the real aim here, is it to link the data to a standard reference point?

A: I would need to know the number of replicates and how what appears to be a single measurement from the root atlas was linked to what I assume was multiple replicates from the rhizobox experiment.

The main discussion/criticism on Root Atlas data is their limitation to single plants at a given stage and a given environment, thereby representing a root system shaped by specific species(genotype)-environment settings which makes generalization questionable. This problem is certainly also valid for the integration of the Root Atlas data with replicated phenotyping measurements in this manuscript. 

Our results (correlations in Fig. 1) indeed points to this issue: length measurements of the single Root Atlas pictures are not correlated to any trait from Rhizoboxes. On the contrary, the Root Atlas values on maximum depth and highest order branching, representing what we called “expert knowledge”, i.e. the outcome from a (unspecified) higher number of observations of the same species in different environments, were correlated to measurement data from Rhizobox plants. This suggests that they represent more generalizable descriptors with can provide added knowledge from integration into other datasets. We discussed this limitation of the Root Atlas data in lines 244 ff.

Application of the multivariate classification method (PCA and clustering) uses single values (i.e. in case of the replicated rhizobox data averages with n=3; in case of the Atlas data, a single number with n=1). Rather than the multivariate analysis as such, it is thus mainly a question whether the trait values used for classification are representative for the population average, which depends on the probability distribution of the trait and the number of replicates to estimate it. The single images from Root Atlas series do not allow any inference on the contribution of genotype/species vs. the specific environmental influence on the trait expression. Values on maximum depth and branching order, considered “expert knowledge” derived from an unspecified number of in-field observation, appear more appropriate as species characteristic, without however being accessible to any variance-based statistics.

The question of standard reference point it not easily answered. For root systems, we indeed cannot expect a strictly family-based delineation (line 345, following the revised discussion on clustering results), but a more continuous variation beyond family limits – although our results demonstrate a certain family-specificity (clustering of legumes and brassicas; revised Fig. 2E and F). We rather assume a functional delineation driven by evolutionary ecosystem adaptation (see e.g. Ma et al. 2018. Evolutionary history resolves global organization of root functional traits. Nature, 555). In this respect, the empirical, data-driven classification approach just provides a statistical step towards identification of distinct root system types, while application of mechanistic root models to the single trait-derived clusters would allow a quantitative verification of a functional background. 

I would like to see a PC plot and dendogram of the Root Atlas data alone, to see how closely these relate to the rhizobox, or atleast the rhizobox positions data points marked separately from the Root Altlas data on the PC plots.  This leads onto the question of whether the clustering in the dendogram sample groups are grouped like that because the rhixobox data is driving the separation  (in Fig 3 b because the rhizobox is driving that data split.

A: Following this important suggestion, we now included the grouping with Atlas data alone (Figure 2A and D) and discussed the results concerning changing in clustering of species (lines 322-340).

 Line 52 – “rely” rather than “relay”

Corrected

Table 2, why is V.faba Underlined and in bold?

Corrected

Table 2 Change the a and b reference links in the titles to symbols

Corrected

Line 108 (CV 103.3 %)  CV should be given in a consistent format see Line88

Corrected

Line 180 “joined” rather than “joint” (and Line 182)

Corrected

Line 250 “angle” not “angel”

Corrected

Did the authors check the likely hood of getting significant correlations by chance when doing the multiple correlations?

A: For the univariate correlations in Fig. 1 among all variables (with trend lines included for correlations with p<0.05) significance tests for the “H0: correlation =  0” were done by a chi-square goodness-of-fit test. Information added in line 490.

Line 387 – 392 – Number of replicates?

Added – line 435.

Line 397, please state resolution of images used for this study.

Added – line 446.

Line 417 – Section Root Atlas – So if I understand this correctly, for each of the root atlas species, there is effectively a sample of 1.

Yes. See comment above concerning the evaluation method.

Why was the dataset of cover crops from ref 30 also not used in this study?

A: The sample (from a previous field experiment) unfortunately did not contain all the species we evaluated here and would have further reduced our combined dataset (including Atlas data where T. alexandrinum is not described) to six species only.

Round 2

Reviewer 1 Report

Accepted all revisions/justifications by the authors.

Reviewer 3 Report

While this study still has limited replication, this is now clearer in the text, and the appropriate limitations have been taken into account.  The authors have now answered all my queries and included the extra figures.